



# Comparison of CO₂ and O₂ fluxes demonstrate retention of respired CO₂ in tree stems from a range of tree species

Boaz Hilman[1], Jan Muhr[2], Susan E. Trumbore[2], Mariah S. Carbone[3], Päivi Yuval[4,5], S. Joseph Wright[6], Gerardo Moreno[7], Oscar Pérez –Priego[8], Mirco Migliavacca[8], Arnaud Carrara[9], José M. Grünzweig[4], Yagil Osem[5], Tal Weiner[1], Alon Angert[1]

[1]The Fredy and Nadine Herrmann Institute of Earth Sciences, The Hebrew University of Jerusalem, Jerusalem, 91940, Israel
[2]Department of Biogeochemical Processes, Max-Planck Institute for Biogeochemistry, Jena, 07745, Germany
[3]Center for Ecosystem Science and Society, Northern Arizona University, Flagstaff, AZ 86011, USA
[4]Robert H. Smith Institute of Plant Sciences and Genetics in Agriculture, Robert H. Smith Faculty of Agriculture, Food and Environment, The Hebrew University of Jerusalem, Rehovot, 76100, Israel
[5]Institute of Plant Sciences, Agricultural Research Organization, Volcani Center, Bet Dagan, 50250, Israel
[6]Smithsonian Tropical Research Institute, Balboa, Apartado 0843–03092, Panama
[7]Institute for Dehesa Research, University of Extremadura, Plasencia, 10600, Spain
[8]Department of Biogeochemical Integration, Max Planck Institute for Biogeochemistry, Jena, 07745, Germany
[9]Instituto Universitario Fundación Centro de Estudios Ambientales del Mediterráneo (CEAM-UMH), Paterna, 46980, Spain

*Correspondence to*: Boaz Hilman (boaz.hilman@gmail.com)

**Abstract.** The ratio of CO₂ efflux to O₂ influx (ARQ, apparent respiratory quotient) in tree stems is expected to be 1.0 for carbohydrates, the main substrate supporting stem respiration. In previous studies of stem fluxes, ARQ values below 1.0 were observed and hypothesized to indicate retention of respired carbon within the stem. Here, we demonstrate that stem ARQ <1.0 values are common across 85 tropical, temperate, and Mediterranean forest trees from 9 different species. Mean ARQ values per species per site ranged from 0.39 to 0.78, with an overall mean of 0.59. Assuming that O₂ uptake provides a measure of *in situ* stem respiration (due to the low solubility of O₂), the overall mean indicates that on average 41% of CO₂ respired in stems is not emitted from the local stem surface. The instantaneous ARQ did not vary with sap flow. ARQ values of incubated stem cores were similar to those measured in stem chambers on intact trees. We therefore conclude that dissolution of CO₂ in the xylem sap and transport away from the site of respiration cannot explain the low ARQ values. We suggest to examine refixation of respired CO₂ in biosynthesis reactions as possible mechanism for low ARQ values.

## 1 Introduction

The global annual CO₂ efflux from tree stems to the atmosphere is estimated at 6.7 ±1.1 Pg C yr⁻¹ (Yang et al., 2016), but the drivers of stem CO₂ efflux are not well understood (Trumbore et al., 2013). CO₂ in tree stems originates primarily from aerobic respiration, which consumes oxygen (O₂). Respiratory quotient (RQ) is defined as the ratio between produced CO₂ and consumed O₂, and its value is derived from the metabolized substrate. Carbohydrates are believed to be the main respiratory substrate in tree stems (Hoch et al., 2003; Plaxton and Podestá, 2006), and their metabolism results in an RQ of ~1.0. Metabolism that relies entirely on lipids yields an



RQ value of ~0.7, but significant storage of lipids in stems is uncommon and limited to several tree genera called
'fat-trees' (Sinnott, 1918). RQ values greater than 1.0 are associated with organic acid catabolism.
Initial measurements of the ratio of $CO_2$ efflux to $O_2$ influx from the stem surface for six tree species found values
mostly below 1.0 (the expected value for RQ from carbohydrate metabolism) (Angert and Sherer, 2011; Angert
et al., 2012). The flux ratio is referred to in those studies, and here, as the "apparent" RQ (ARQ), because it
potentially includes processes that incorporate $CO_2$ and/or $O_2$ in the stem in addition to the respiration taking place
in tissue beneath a chamber placed on the stem surface. Processes that can potentially reduce the emission of $CO_2$
and thereby decrease ARQ below 1.0 include dissolution and transport of $CO_2$ in the xylem sap (Teskey et al.,
2008), and carboxylating reactions during biosynthesis of compounds more oxidized than carbohydrates
(Lambers et al., 2008). Alternatively, it may be hypothesized that ARQ below 1.0 is the result of non-respiratory
$O_2$ uptake, e.g. by oxidases and hydroxylases that are $O_2$ consuming enzymes.
Carbon dioxide is ~30 times more soluble in water than $O_2$, and dissolved $CO_2$ reacts with water to form
bicarbonate ($HCO_3^-$) and carbonate ($CO_3^{2-}$) ions, further increasing the amount of dissolved inorganic carbon
(DIC). The rate of $O_2$ uptake is thus assumed to provide a better measure of stem respiration than $CO_2$ efflux,
which can be complicated by dissolution and transport within the xylem sap (Teskey et al., 2008), potentially
contributing to low ARQ values. If transport of $CO_2$ within the stem is important, ARQ measured at the stem
surface is expected to be inversely related to sap velocity. As the difference in solubility between $CO_2$ and $O_2$
decreases with increasing temperature (Gevantman, 2018), ARQ might be expected to increase with temperature
if all other factors remain constant. In addition, variations of ARQ with stem height are to be expected. A model
of $CO_2$ diffusion and advection in the xylem sap by Hölttä and Kolari (2009) predicted that the accumulation of
dissolved $CO_2$ in the ascending xylem sap, together with a reduction in stem diameter with height, induces faster
$CO_2$ diffusive loss to the atmosphere in the upper parts of the stem. Thus, an increase in ARQ (higher $CO_2$ loss
per mole of $O_2$ uptake) with stem height is expected. However, there is evidence from studies with an isotopically
labeled stem $CO_2$ pool that a significant portion of C is transported as DIC to photosynthetic tissues where it might
be refixed to organic C (Bloemen et al., 2013; McGuire et al., 2009; Powers and Marshall, 2011). To date, studies
of these processes in large trees are scarce, and it is not clear which process are responsible for low ARQ. If lower
than unity ARQ values are prevalent and result from processes that retain $CO_2$ in the stem, estimates of tree stem
respiration based on $CO_2$ efflux measurements must be reconsidered. Thus, the first objective of this work is to
determine whether ARQ values lower than 1.0 is observed in a variety of trees from different biomes and across
seasons. A secondary objective of this study was to test whether ARQ varies with xylem stream characteristics or
with tree height.
**2 Materials and Methods**
**2.1 Methods for evaluating ARQ**
We report tree stem ARQ results based on measurement methods described in (Hilman and Angert, 2016). These
methods overcome the difficulty of measuring small changes in $O_2$ against the high atmospheric background by
using a static stem chamber, in which the $O_2$ changes are considerably larger than in an open flow chamber.
We used three different approaches to measuring ARQ: two are based on discrete gas samples of headspace air,
and one based on direct measurement of instantaneous fluxes using gas sensors in the first hour after chamber





sealing. Discrete gas samples are either taken within 30 minutes to few hours after chamber sealing
("instantaneous" sampling) or after the chamber has been sealed to the stem for more than 24 hours, once steady
state conditions have been achieved ("steady state"). These timings were confirmed by continuous measurements
with sensors (Hilman and Angert, 2016).

### 2.1.1 ARQ measurement from discrete samples

The evaluation of ARQ from discrete gas measurements is based on a one-box model that describes gas dynamics
in the headspace of a static chamber sealed to the surface of a tree stem (Angert and Sherer, 2011; Angert et al.,
2012; Hilman and Angert, 2016). In the model, the gas in the chamber headspace has initial mean atmospheric
values (20.95% $O_2$, 0.04% $CO_2$), ensured by flushing the chamber with ambient air before measurement. Once
the chamber is closed and the headspace above the stem surface is isolated, metabolic reactions in the stem control
the chamber's air composition. For the first few hours, headspace concentrations of $CO_2$ increase and $O_2$ decrease
at rates that are roughly linear with time ("instantaneous" incubation, Fig. 1,S1). During this linear stage, ARQ is
calculated by:
$$ARQ = \frac{CO_2 \text{ efflux}}{O_2 \text{ influx}} = \frac{\Delta CO_2}{\Delta O_2} \qquad (1)$$
where $\Delta CO_2$ and $\Delta O_2$ are the changes in $[CO_2]$ and $[O_2]$ during the initial period after the chamber was sealed,
and for discrete samples can also be determined from the difference in concentrations between the chamber air
sampled at a specific time and the initial atmosphere. "Instantaneous" fluxes of $CO_2$ and $O_2$ reported here are
obtained either by monitoring concentration change during the first hour following chamber closure with sensors
directly in the field or by sampling headspace air with glass flasks within 30 minutes to a few hours of closing the
chamber. The flasks were transported to the laboratory for measurement of $CO_2$ and $O_2$.
After the first hours, the initially linear rates of change in headspace gas concentration with time decline, and
concentrations eventually remain constant (Fig. 1,S1). In this phase the gases in the chamber and the outer part of
the stem, where most of the metabolism takes place, are assumed to be in equilibrium. This "steady state" occurs
when the rates of addition of $CO_2$ and loss of $O_2$ from the stem to the chamber headspace are balanced by diffusive
(assuming no strong wind) exchange of headspace air with outside air through porous portions of the outer stem.
For "steady state" samples, the chamber is sealed to the surface of the stem and left for a period longer than 24
hours, after which the headspace air is sampled using glass flasks. The $CO_2$ and $O_2$ concentrations must be
corrected for differences in diffusivity between $CO_2$ and $O_2$, as detailed in (Angert and Sherer, 2011; Angert et
al., 2012; Hilman and Angert, 2016) in order to estimate the ratio of the gas fluxes from the concentrations in the
static chamber:
$$ARQ = \frac{gCO_2 \times \Delta CO_2}{gO_2 \times \Delta O_2} \qquad (2)$$
where $gCO_2$ and $gO_2$ are the $CO_2$ and $O_2$ conductance values in the outer layer of the stem between the chamber
and the atmosphere. The structure of the path along which diffusion occurs is the same for $CO_2$ and $O_2$ and hence
the conductance ratio $gCO_2/gO_2$ depends solely on the ratio of diffusivities of the gases in air, which is 0.76
(Massman, 1998). As a result, at steady state:
$$ARQ = 0.76 \times \frac{\Delta CO_2}{\Delta O_2} \qquad (3)$$
Hilman and Angert (2016) demonstrated excellent agreement for direct comparisons of the "instantaneous" and
"steady state" measurement methods, and the results are further compared here.



The data we report here were collected in different sites and over different years, and chamber designs and
methods applied varied from site to site, as described in Sect. 2.2 and in Table 1. In all cases, a chamber is attached
to the surface of the stem with an air-tight seal (using a sealant in most cases – see Table 1 for details). Ports (to
which sampling flasks can later be attached) or a separate lid with ports allow the chamber to remain open to the
atmosphere when not in use; openings are covered with screen to prevent insect damage inside the chambers. For
a measurement, the chamber is first flushed with ambient air using a syringe, then all openings are closed, and
$CO_2$ is allowed to accumulate (and $O_2$ to be consumed) in the headspace trapped within the chamber. The
chambers contain sampling ports to which glass flasks equipped with O-ring valves (LouwersHanique, Hapert,
The Netherlands) are attached. Initially the valves are open. Air from the chambers is sampled passively by closing
the valves. For "steady state" field measurements, two glass flasks are connected to a stem chamber and closed
after at least one day of incubation. For "instantaneous" ARQ, the valves are closed after shorter incubation
periods (30 minutes to a few hours).
The flasks were analyzed in the laboratory at the Hebrew University in Jerusalem in a closed system [The
*Hampadah* (Hilman and Angert, 2016)]. Two analyzers are included in the *Hampadah* system; an infra-red gas
analyzer (IRGA) for $CO_2$ measurement (LI 840A LI-COR; Lincoln, NE, USA) and a fuel-cell based analyzer
(FC-10; Sable Systems International, Las Vegas, NV, USA) for measuring $O_2$. The principle of operation of the
*Hampadah* is measurement of the change in $CO_2$ and $O_2$ concentrations in the system's air after flask opening,
and calculation of the concentration in the flask that would yield such change.

### 2.1.2 Continuous ARQ measurements

Sensitive detection of small changes in $O_2$ is difficult in the field, which is why we used the flask samples and
long chamber closure times ("steady state") in most field sites. However, to measure diurnal changes in stem ARQ
values of *Malus domestica*, we were able to make continuous measurements with a small IRGA $CO_2$ sensor
(COZIR Wide Range 0-20% $CO_2$ Sensor, CO2Meter, Inc.) and a quenching based optode (Fibox 3, PreSens-
Precision Sensing) for $O_2$ measurement (Hilman and Angert, 2016). The sensors' reading was extracted every 30
seconds. A temperature sensor was placed next to the optode sensor for temperature and water vapor corrections.
The inlet of a small diaphragm pump (KNF micro-pump) and a non-return valve (SMC AKH 12mm, RS, UK)
were connected to the chamber headspace, for periodic automatic venting of the chamber every 4 hours. The $CO_2$
efflux and the $O_2$ influx were calculated using a linear fit over ~120 gas concentration measurements during the
first hour of incubation, the chamber volume, and the stem surface area under the chamber. We used the data from
this experiment to examine the sensitivity of ARQ to temperature, which affects the gas solubility constants. The
strongest effects are expected during the night, when daytime influences on stem fluxes associated with sap flow
and low turgor pressure (Salomón et al., 2018) are minimized.
For each site and experiment described below, we identify the method used to estimate ARQ as
"instantaneous", "steady state" (for flask samples) or "continuous".

### 2.2 Study sites and experimental design

For addressing our first goal of determining the variation in stem ARQ values across a range of tree species and
environments, we measured ARQ in trees located in tropical forests in the Republic of Panama and in Brazil, in
temperate forests in the northeast US (Bartlett and Harvard forests), in Mediterranean savanna in Spain, and in





Israel where we sampled various species on the Hebrew University campus in Jerusalem and the adjacent Botanical Gardens, and in natural Mediterranean shrubland that is located on the Carmel Ridge (Table 1, Fig. 3). Seasonal measurements were performed in Jerusalem, US, and Brazil sites. In Jerusalem, five individual trees from five different species (first five species in Table 1) were measured every 2-3 months between December-February 2011 and July 2014, except for the *M. domestica*, which was measured at monthly intervals between July 2011 and July 2013 ("steady state"). Phenology of the deciduous trees (all except *Quercus calliprinos*) was classified into four groups (Fig. 4). In addition, in the same site, we sampled four *Quercus ilex* trees in July 2016 ("steady state"). Five individuals of *Acer rubrum* were measured at each of the sites in the US in September 2012 ("steady state"). Trees at the northern site (Bartlett Experimental Forest) had fall color development, while leaves at Harvard Forest were still green. We questioned if ARQ would vary with the phenological differences. After analysis we excluded results from three trees because of suspected air leakage from the chamber ($O_2$ >20% after six days of stem incubation). In Brazil six *Scleronema micranthum* trees were measured in five campaigns between March 2012 and March 2014. In the two first campaigns "instantaneous" ARQ was measured, while "steady state" ARQ was measured in the three later campaigns. After analysis we excluded results from four measurements because of a weak signal ($O_2$ >20.7% and SD >0.1 after 3 h of incubation). In Panama we sampled 42 *Tetragastris panamensis* trees ("steady state") in three campaigns: September 2012, September-October 2013, March-April 2014. Some individuals were sampled more than once. The trees grew in plots that were part of a fertilization and litter manipulation projects (Wright et al., 2011; Sayer and Tanner, 2010). No treatment effects were found (Fig. S2). In Spain we sampled 16 *Q. ilex* trees during May 2015 ("steady state"). On Carmel Ridge we sampled ARQ of four *Q. calliprinos* trees ("steady state") during April 2012, September 2012, and January 2013.

For our second objective, to explore the potential for low ARQ values to reflect dissolution and transport of $CO_2$ in the xylem sap, we measured instantaneous ARQ at varying sap flow velocities and in different times of a day. Transport of $CO_2$ was previously reported to be correlated with sap flow (McGuire and Teskey, 2004; Bowman et al., 2005; McGuire et al., 2007). Thus, anti-correlation of ARQ with sap flux, expressed in maximal values during night in diel course, would provide evidence to support the transport of locally respired C as DIC. We also measured vertical transects of ARQ including in-stem measurements, using chambers and probes placed at different heights on a single stem, and in incubations from stem cores. We performed a number of experiments:
1) ARQ ("instantaneous") was measured simultaneously with sap flux density measurements in nine *Q. ilex* trees with similar diameter (0.35 to 0.49 m at breast height) in the site in Spain.
2) At sites where we could not measure sap flux, we measured day-night variation in ARQ. ARQ ("instantaneous") was measured during daytime, at pre-dawn when the transpiration stream should reach its minimum, and again during the next day. We conducted two day-night campaigns on the trees at the site in Jerusalem, during July 2012 and April 2013. Additionally, during 24-28 April 2013, ARQ values were measured every 4 h from the *M. domestica* tree in Jerusalem ("continuous").
3) ARQ ("steady state") was measured over spring, summer and winter in the *Q. calliprinos* trees on Carmel Ridge site, simultaneously with pre-dawn shoot water potential ($\Psi_{pd}$). $\Psi_{pd}$ is a measure for available soil water and therefore is also a rough proxy for seasonal differences in transpiration rates (Aranda et al., 2005; Bucci et al., 2005).





4) ARQ was measured at different heights on the same tree stem from the stem surface using stem chambers, and
also from inside the stem. In the ARQ seasonal measurements in Jerusalem, the *Q. calliprinos* and the *Platanus*
*occidentalis* trees were measured at their stem base, in addition to the breast height measurement ("steady state").
In Brazil, we measured "instantaneous" ARQ from stem chambers installed up to 11 m above the ground on a
single *S. micranthum* tree. To evaluate the influence of the internal ARQ on the surface ARQ, we measured in the
same tree in-stem gas concentrations and ARQ.
5) "Steady state" ARQ measured from stem chambers was compared with ARQ measurements through incubation
of stem cores. Measurement of stem tissues should provide better estimation for the stem RQ by excluding
dissolution and advection in the xylem stream. Incubations were performed on cores taken from four species in
four different sites (Table 1). In Jerusalem, we compared stem incubation ARQ with that of leaf incubation.

### 2.3 Sap flux density

Sap flux density was monitored in 9 trees at the site in Majadas de Tietar (Spain) using heat ratio method (HRM)
sensors (SFM1 Sap Flow Meter, ICT International). A description of the installation and measurement is presented
in Methods S1. The detailed procedures for sap flux corrections and calculations are described in (Perez-Priego
et al., 2017). We tested, whether the daily maximum sap flux density (i.e. average of measurements between 10:00
and 17:00 during the day of the ARQ measurement), which correlated with $CO_2$ dissolution fluxes (Bowman et
al., 2005), would explain variability in "instantaneous" ARQ.

### 2.4 Shoot water potential

Pre-dawn shoot water potential ($\Psi_{pd}$) on Carmel Ridge was measured using a pressure chamber (PMS Instrument
Company, Corvallis, Oregon, USA). At each sampling time, we sampled 2-3 terminal twigs containing 5-10
leaves from each *Q. calliprinos* tree. The samples were wrapped in plastic, placed on ice and measured within an
hour of sampling using the pressure chamber technique (Scholander et al., 1965).

### 2.5 In-stem measurements

For sampling gas from inside the stem, stainless-steel tubes (1.3 cm diameter) were installed 4, 8, and 12 cm deep
into the stem, in various stem heights on the same *S. micranthum* tree in Brazil where the vertical ARQ transects
were measured. Installation procedure was according to Muhr et al. (2013) and tubes were sealed between
sampling dates. Using rubber tubing we connected the sampling flasks to the tubes for incubation of 4 days. The
flasks were then analyzed for $CO_2$ and $O_2$ in the *Hampadah*. Assuming steady state, ARQ was calculated using
Eq (3) (Angert et al., 2012).

### 2.6 Measuring ARQ of incubated tissues

Stem cores were extracted in Panama, Spain, and Jerusalem using 1.2 cm diameter cork borer, right after the
chamber incubation experiment. The outer bark and green tissues, as well as sapwood sieves (with paler color
than the phloem tissues), were removed from the cores. The cores were cut to fit into the incubation system,
wrapped with moist gauze cloth to avoid desiccation, and inserted into gas-tight set of flasks (two or three)
connected by Swagelok Ultra-Torr fittings (Swagelok, Solon, OH, USA, Fig. S3). At the end of the incubation





period, the flasks were closed and analyzed in the *Hampadah*. Since the incubations took place in a closed system,
the change with time in [$CO_2$] and [$O_2$] are assumed to be linear, and ARQ can be calculated using Eq (1).
In Panama and Spain the incubations started immediately upon core extraction, at ambient temperature, and lasted
8 h and 3 h, respectively. In Israel, the *Q. ilex* cores were kept on moist gauze cloth for 2 h before being sealed in
the incubation system that were kept at 25°C in environmental chamber. Repeated incubations were performed in
series, with the incubation systems flushed in between with ambient air. Simultaneously, from each tree, four
leaves from an understory branch were cut and inserted into the same incubation systems, for the same incubation
durations. The $O_2$ uptake rate (nmol $O_2$ g.FW$^{-1}$ s$^{-1}$) was calculated as follows [adopted from Pruyn et al. (2002a)]:
$$O_2 \text{ uptake rate } = \frac{\Delta O_2}{100} \times \frac{V_H}{T \times M_{FW} \times V_m} \times 10^9 \qquad\qquad (4)$$

where $\Delta O_2$ is the decrease in [$O_2$] during the incubation, $V_H$ is volume of headspace (ml), T is incubation period
(s), $M_{FW}$ is fresh weight (g), $V_m$ is the molar volume, and $10^9$ converts units to nmol.

239         In Brazil, stem cores were extracted by using 5.15 mm increment corer. After bark was removed the

cores were cut to a length of 6 cm and then allowed to equilibrate with the atmosphere for 6-8 hours, while
continually being kept moist. After equilibration, each core was transferred to an incubation chamber equipped
with flasks. Prior to starting the incubation, a few ml of water were added to keep the core tissue moist. In this
case, incubations were left at room temperature for 24 h before flasks were closed and removed.

## 2.7 Statistical analysis

All statistical analysis was done using JMP (JMP®, JMP Pro 13, SAS Institute Inc., Cary, NC, USA). Repeated
measures analysis of variance was used to evaluate how the interaction of tissue (stem core/leaves) with ARQ and
$O_2$ uptake varies with time in the repeated incubations of the *Q. ilex* tissues. Mauchly's test indicated violation of
sphericity in the ARQ response in the repeated incubations experiment ($\chi^2$ =18.132, $P$ =0.021), therefore
Greenhouse-Geisser adjusted F test was chosen. One-way analysis of variance (ANOVA) followed by Tukey-
Kramer HSD was used to perform comparisons among time points in every tissue. Student's t-test was used for
comparisons between stem cores and leaves at each time point.

## 3 Results

The ARQ estimated from "instantaneous" and "steady state" measurements were in good agreement over a large
range of ARQ (Fig. 2). The average "steady state" ARQ value across all species and sites, including results from
(Angert et al., 2012), was 0.59 (n =229) and the average ARQ of species in the different sites ranged between
0.39 and 0.78 (Fig. 3). For individual measurements, a minimum ARQ value of 0.27 was recorded for *Q. ilex* in
Spain and for *T. panamensis*. The highest value was 0.99 for *M. domestica* and *Populus deltoids*.
Phenology or seasonality had some effect on ARQ (Figure 4). In Brazil, ARQ varied between 0.41 ±0.15 in the
wetter season and 0.82 ±0.12) in the drier season. In Jerusalem, the ARQ of *Q. calliprinos* and *Pistacia atlantica*
had lowest values during spring and highest values in fall and winter (Fig. 4). The average ARQ of the *A. rubrum*
trees at Harvard Forest, where all leaves were green, was significantly greater than the average ARQ of the trees
at Bartlett Experimental Forest, where the leaves had autumn color development (0.69 vs. 0.57, P <0.05 in a
Student's t test).





**3.1 ARQ values under varying xylem stream flow and temperature**
Instantaneous ARQ values of nine *Q. ilex* trees were invariable (mean ±SD of 0.42 ±0.04) in comparison with the
larger variation in maximum daily sap flux density among these trees (0.15 ±0.05 m$^3$ H$_2$O m$^{-2}$ h$^{-1}$), and no
correlation was found between the ARQ and sap flux density (r$^2$ =0, $P$ =0.9891).
Mean ARQ ±SD values ("steady state") of the oaks at the Carmel Ridge site were 0.62 ±0.06, 0.68 ±0.07 and 0.69
±0.08 for spring, summer and winter, respectively. Repeated-measures analysis of variance found no significant
difference between seasons (F$_{2,2}$ =2.52, $P$ =0.28), while Ψ$_{pd}$ varied significantly with seasons (F$_{2,2}$ =207.85, $P$
=0.0048). During summer, Ψ$_{pd}$ was -2.65 MPa, much lower than the spring and winter values (-0.64 and -0.86
MPa, respectively).
In the day-night campaigns done at Hebrew University and the adjacent arboretum, ARQ
("instantaneous") values ranged between 0.52 and 1.05, across all trees, seasons, and sample times (Fig.5). Pre-
dawn ARQ values higher than daylight values (beyond the duplicates error) were observed during the summer in
*M. domestica* and in the upper chamber on *Q. calliprinos*. No significant diurnal effect was found in repeated-
measures analysis of variance of the breast height chambers, neither when results of all the trees was grouped by
season, nor when results were grouped by stem chamber. In "continuous" measurement of *M. domestica*, ARQ
value obtained every 4 hours. ARQ during night was not significantly higher than the day time ARQ value [$P$
>0.76 in a student's $t$ test, 0.70 (n =12) vs 0.71 (n =11) respectively, Fig. 6]. The variations among the nighttime
values were best fitted with temperatures measured 235 minutes before measurement (r$^2$ =0.84, $P$ =0.0001, ARQ
=0.01 × Temperature (C°) + 0.54). With the same time lag, the coefficient of determination for the daytime values
is r$^2$ =0.44 ($P$ =0.0266).

**3.2 Stem surface and in-stem ARQ vertical transects**
In *Q. calliprinos,* measured over three years, ARQ did not differ significantly ($P$ >0.33 in student's $t$ test) between
breast height and stem base (ARQ of 0.56 vs. 0.59 respectively, n =14, Fig. 4). For *P. occidentalis* measured for
the same period the ARQ measured at breast height was significantly higher than ARQ measured at the stem base
(0.74 vs. 0.64 respectively, n =12, $P$ =0.003 in student's $t$ test, Fig. 4). For a single *S. micranthum* tree in Brazil,
ARQ values measured at heights of 6.5 m and 11 m above the ground were similar to ARQ measured at breast
height (Fig. 7), but also show differences with the stem base. In this tree, ARQ measured in March (0.46 ±0.11)
was lower than in October (0.89 ±0.16). The in-stem ARQ values ranged between 0.25 and 0.56, with average
±SD of 0.46 ±0.07 in both seasons and at all stem positions and depths. The in-stem ARQ, as well as [CO$_2$] values,
had no clear vertical trend (Fig. 7,S4).
**3.3 Tissue incubations**
The average ARQ values of the stem core incubations were similar to the stem chamber ARQ for the four
sites/trees where these comparisons were made (Fig. 8). In the time series incubations of *Q. ilex* stem cores and
leaves, significant effects of time, tissue (leaves, stem cores), and their interactions (time × tissue) on ARQ and
O$_2$ uptake rates were observed. ARQ of the stem cores increased from 0.44 ±0.08 (mean ±SD, n =4) after 3 h to
0.94 ±0.03 at the end of the experiment (32 h; Fig. 9). The ARQ of incubated leaves of the same trees showed
higher initial ARQ of 0.80 ±0.02, with an increase over time to 0.92 ±0.02.



## 4 Discussion

### 4.1 ARQ is lower than 1.0 for a wide range of tree species

The ARQ measured in stem chambers installed on 85 individual trees of 9 species including tropical, temperate and Mediterranean forest trees was considerably and almost universally lower than 1.0. ARQ values as low as 0.7 could indicate that lipids were used exclusively as substrates for respiration. Lipids respiration is often associated with environmental stresses, for example, initial RQ of ~1 measured in branches of *Pinus sylvestris* L. declined in response to 11 days of shading and drought treatments to values of 0.77-0.75, reflecting mixture of substrates (Hanf et al., 2015). However, many ARQ values are below 0.7, so substrate use alone cannot explain them. Additionally, as ARQ values above 1.0 are expected when lipids are produced (De Vries et al., 1974), ARQ <1.0 resulting from lipid metabolism must be mirrored with ARQ >1.0 at a different time (assuming the lipids are produced locally). However, ARQ almost never exceeded 1.0. The results demonstrate that $O_2$ influx to the stems usually exceeded the $CO_2$ efflux, regardless of tree species, site, season, and time of day. Assuming $O_2$ uptake provides a measure of *in situ* respiration (due to the low solubility of $O_2$), values of ARQ averaging 0.59 indicate that on average 41% of the $CO_2$ produced by respiration was not locally emitted to the atmosphere, but apparently retained in the stem. For sites where we have time series data for the same individuals, considerable variation in ARQ values was observed over two years in Brazil and over three years in Israel. A decrease in ARQ values was often observed during entrance to dormancy for the deciduous trees in Jerusalem, and an apparent minimum in ARQ for *P. atlantica* and *Q. calliprinos* in spring (Fig. 4). This seems to be in agreement with the findings of significantly lower ARQ for Bartlett forest, where leaves were beginning to senesce, compared to the more southerly Harvard forest, where leaves were still green.

The possibility of measurement artifacts as the source for the low ARQ values seems unlikely, as Hilman and Angert (2016) demonstrated the validity of the measurement methods and the box-model approach. Further support comes from the slope (1.006) of the linear regression of "instantaneous" ARQ vs. "steady state" ARQ measured for the same tree, which is extremely close to 1 (Fig. 2). The considerable scattering in the regression may be attributed to temporal differences in the time integrated by the two types of measurement: the "instantaneous" sampling was typically conducted few days before the "steady state" sampling on the same tree, and to lower precision in "instantaneous" samples due to smaller changes in $O_2$ over the shorter time periods (Hilman and Angert, 2016). We also found strong similarities between the ARQ measured for intact stems with chambers and by incubating cores (Fig. 8), which provides another, indirect, confirmation that the low ARQ values obtained with the stem chamber measurement approaches are measuring something that is occurring in the stem tissues.

### 4.2 Dissolution and transport of respired $CO_2$ in xylem stream cannot explain the low ARQ values

Given the low solubility of $O_2$, stem flux ARQ values <1.0 (or potentially 0.7 for 'fat' trees) are either result of respired $CO_2$ that is exported from the site of respiration before it can be emitted to the atmosphere or refixed in biosynthesis processes. As noted earlier, a second possibility is non-respiratory $O_2$ uptake, e.g. by oxidases and hydroxylases that are $O_2$ consuming enzymes, most notably used in lignin biosynthesis. However, stoichiometric analysis of this pathway shows that the $CO_2$ produced from the sucrose that is the lignin's substrate usually exceeds the $O_2$ consumption, so that the net effect of lignin biosynthesis should be a local increase in ARQ (Amthor,





2003). To the best of our knowledge, there are no other significant $O_2$ consuming processes in tree stems that
might affect the ARQ value.

342         We conclude that the low ARQ much be the result of $CO_2$ being locally fixed or transported away from

the site of respiration. If $CO_2$ dissolution and DIC transport is the main export mechanism, we would expect ARQ
to be related to temperature (i.e. according to solubility changes with temperature), anti-correlated with sap flow
(McGuire and Teskey, 2004; McGuire et al., 2007; Bowman et al., 2005), and further that ARQ should increase
with height in the stem (Hölttä and Kolari, 2009). Three observations support the idea that this export mechanism
controls some of the variability in ARQ. First, in the diurnal measurements of the *M. domestica*, the nighttime
ARQ results were indeed correlated with temperature, an expected trend given the greater temperature sensitivity
of the $CO_2$ solubility in comparison with $O_2$ (Gevantman, 2018). Second, the *P. occidentalis* had higher ARQ
values in the upper stem position, especially during the growing seasons (Fig. 4). Third, relatively high ARQ
values were observed at 0.2 m above the ground in the *S. micranthum* tree (Fig. 7), which may reflect a burst of
in-stem $CO_2$ that originated from belowground respiration (McGuire and Teskey, 2004; Levy et al., 1999).
However, in most of our observations ARQ did not vary as expected if $CO_2$ dissolution and transport were the
main $CO_2$ export mechanism.

355         When sap flux density was measured directly, it did not explain the variation in ARQ among *Q. ilex* trees

in Spain. Mean ARQ values were fairly stable over spring, summer and winter (0.62-0.69) for *Q. calliprinos* in
the Carmel Ridge site, while the transpiration stream probably varied greatly between seasons if related to $\Psi_{pd}$.
Additionally, during dormancy when no leaves were in place to force the transpiration stream, we found ARQ
values <1.0 in four deciduous trees (black markers in Fig. 4). Transpiration streams are also assumed to decline
during the night, but ARQ values <1.0 during nighttime were measured in five species, and in most cases no
nocturnal increase of ARQ in comparison to daytime values was observed (Fig. 5,6). Thus, the temperature
dependency observed for the *M. domestica* tree during the night, which explained variability in ARQ values
between 0.65-0.75, must be a second order control on ARQ variability and cannot explain the big deviation from
unity (according to the linear fit, an ARQ of 1.0 is expected at the unreasonable temperature of 63°C). Also, the
vertical transects of ARQ for *Q. calliprinos* and *S. micranthum*, including in-stem ARQ for the later (Fig. 4,7,S4),
showed no consistent pattern of ARQ increasing with stem height, unlike the ARQ increase with height measured
in the *P. occidentalis* (Fig. 4). Likewise, no trend of in-stem [$CO_2$] increase with stem height was observed,
suggesting no $CO_2$ accumulation in the ascending xylem sap (Fig. S4).

369         The in-stem ARQ measured in the *S. micranthum* ranged between 0.25 and 0.56. Even lower values,

with typical ARQ of 0.13-0.18, have been reported before (Angert et al., 2012). These low values are consistent
with in-stem measurements of small $CO_2$ increases and large $O_2$ reductions in comparison to atmospheric
concentrations (Pruyn et al., 2002b; Eklund, 1990; Eklund, 1993). Thus, one can speculate that the low ARQ
values at the trunk surface are the reflection of the low ARQ in the sapwood itself. However, there are
contradicting assessments about the influence of in-stem $CO_2$ on the $CO_2$ efflux from the stem surface; while
some studies interpreted tight covariations of in-stem $CO_2$ and surface efflux as strong in-stem influence (Teskey
and McGuire, 2007; Steppe et al., 2007; Teskey and McGuire, 2002), other studies inferred only marginal
influence of in-stem processes on surface efflux (Ubierna et al., 2009; Maier and Clinton, 2006). Unlike
covariation observations, which do not necessarily represent cause-and-effect relationships (Maier and Clinton,
2006), Muhr et al. (2013) utilized the difference in $^{14}C$ signature of in-stem $CO_2$ (5 cm deep) and surface efflux



to estimate that <20% of total emitted $CO_2$ originates from the inner stem. To assess the potential influence on
ARQ measured at the surface, we used a two pool model for sources of the surface $CO_2$ efflux: (1) an in-stem
$CO_2$ pool that is affected by sap flow transport, and (2) $CO_2$ produced locally by stem tissues (mostly close to the
stem surface) fully released to the atmosphere. Using the results of the *S. micranthum*, we can evaluate the
contribution of the in-stem ARQ to the stem surface ARQ. The mean ARQ values for the 4 cm deep probes and
for the surface were ~0.5 and 0.67, respectively. Assuming 20% of the $CO_2$ efflux comes from the in-stem due
the diffusive gradient between the sapwood and the atmosphere, the other 80% comes from metabolism in the
stem tissues close to the stem surface. With an in-stem ARQ of 0.5, which means that the $O_2$ influx induced by
concentration gradient is twice the $CO_2$ flux after correcting for relative diffusivity, the $O_2$ influx between the
atmosphere and the inner-stem would be equivalent to 40% of the $CO_2$ efflux. To explain the overall ARQ
(measured at the stem surface) of 0.67, the $O_2$ influx to the stem tissues must therefore be the equivalent of 110%
of the $CO_2$ efflux, i.e. the total ARQ representing the sum of fluxes from diffusion and metabolism would be
(20+80)/(40+110) =0.67. The ARQ of the fluxes stem tissue metabolism alone would be 80/110 =0.73, which is
still lower than unity.
ARQ values <1.0 observed in stem core incubations, where tissues are isolated from the influence of
transport (Fig. 8,9) further support our conclusion that ARQ resulting from local metabolism of the stem tissue
near the surface is <1.0. In light of that, the apparent decoupling between ARQ, sap flux density, and $\Psi_{pd}$ presented
above might be derived from strong diffusion barriers that restrict gas exchange between the sapwood and the
external cambium, phloem, and bark tissues (Ubierna et al., 2009). A major contribution from respiratory activity
concentrated in the outer stem tissues to overall stem respiration would further reduce sap flow effects on surface
fluxes (Hölttä and Kolari, 2009; Maier and Clinton, 2006; Ubierna et al., 2009).
Overall, our results suggest that $CO_2$ dissolution and removal in the xylem stream are not the main cause of the
low ARQ values that are common to the trees we measured. At the same time, observed ARQ values may be
influenced by the cumulative effects of some dissolution and transport, partial lipid metabolism, and non-
respiratory $O_2$ consumption. One potential explanation for low ARQ values could be biosynthesis with
engagement of the enzyme phosphoenolpyruvate carboxylase (PEPC), which is able to fix respired $CO_2$. Indirect
evidence for PEPC activity can be found in the increase of the ARQ values with time in our repeated incubations,
while cellular activity was retained as reflected in $O_2$ uptake rates (Fig. 9). Such a pattern may reflect a biochemical
process, e.g. C fixation by the enzyme PEPC, that decreases with time due to self-inhibition by the accumulation
of the products (Kai et al., 1999; Huber and Edwards, 1975). Based on mass balance calculation for the stem cores
incubations and published PEPC fixation rates in tree stems (Table 2), PEPC fixation rates can easily explain the
retained $CO_2$. The fixation products, organic and amino acids, may be exported via the xylem stream and/or via
the phloem (Hoffland et al., 1992; Schill et al., 1996). Further investigation into the potential role of PEPC,
including direct measurement of PEPC activity, would be needed to assess whether PEPC plays a role in lowering
ARQ values to the levels observed. To complete stem carbon balance, additional evaluation of the relations
between the in-stem and the stem surface fluxes are also needed, as well as analysis of how organic and amino
acids vary in the stem.





### 4.3 Implications of low ARQ

From a whole ecosystem perspective, if respired $CO_2$ in the stem returns to the atmosphere elsewhere (e.g. in the soil, canopy), the overall ecosystem-atmosphere carbon fluxes will not be affected, and high ARQ associated with the release of the transported $CO_2$ will balance the low ARQ in the stem. However, such internal transport can cause a discrepancy between the measured above-ground and below-ground $CO_2$ effluxes and the locations where respiration is actually occurring (Aubrey and Teskey, 2009), and lead to false attribution of respiration responses to environmental conditions. Moreover, the different long-term temperature sensitivity of $CO_2$ efflux and $O_2$ influx is of interest, and might explain part of the gap between modeled and observed $Q_{10}$ values of tree respiration (Griffin and Prager, 2017). For example, decrease in ARQ with rising temperature (due to higher PEPC activity for example) might result in a slow increase in $CO_2$ efflux, whereas the respiration rate ($O_2$ uptake) is actually increasing sharply, together with the internal carbon flux. Future studies should determine how temperature and nutrients control long term changes in ARQ, and aim to identify the biochemical process that control the low ARQ reported by the current study.

*Author Contribution.* B.H and A.A planned and designed the research. B.H performed the ARQ analysis and led the writing of the manuscript. J.M and S.T carried out the field work in Brazil, and M.S.C carried out the field work in USA. P.Y measured shoot water potential. S.J.W designed the long term experiment in the Republic of Panama. G.M, O.P, M.M, and A.C contributed to the campaign in Spain. O.P measured the sap flux density. J.M.G and Y.O contributed to the campaigns in the Carmel Ridge. T.W contributed to the campaigns in Spain and in Givat Ram campus. J.M, S.T, S.J.W, G.M, O.P, M.M, J.M.G and A.A contributed to the discussion and writing.

*Data availability.* Data used in this study can be found in figures, tables and in the Supplement.

*Competing interests.* The authors declare that they have no conflict of interest.

*Acknowledgements.* This research was funded by the German-Israeli Foundation for Scientific and Development (#1334 /2016), and by a Ring Foundation grant. B.H was partly funded by the Advanced School for Environmental Studies and by the Canadian Friends of the Hebrew University. MM and OPP acknowledge the Alexander von Humboldt Foundation for supporting the research activities of the MANIP project with the Max-Planck Prize to Markus Reichstein. We thank Avihay Berry for assistance with field work at Jerusalem, Itsik Simchayov for technical support, Ramat Hanadiv Nature Park for logistic assistance and the Jerusalem Botanical Gardens in Givat-Ram and especially Dr. Ori Fragman-Sapir and Ofri Bar for facilitating this research, and Rufino Gonzales and Omar Hernandez for assistance with field work at Barro Colorado Nature Monument.

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



**Table 1** Study sites, tree species sampled at each site, stem chambers and stems dimensions.

| Site and coordinates | Species | Chamber type, sealant | Diameter at the base of the chamber (cm)[a] | | | Measurements made in the site | ARQ measurement method |
|---|---|---|---|---|---|---|---|
| | | | Mean ± SD | minimum | maximum | | |
| Givat Ram campus, Jerusalem, Israel (31.77°N, 35.20°E) | *Populus deltoids* Bartr. Ex Marsh | Perspex®[b], hot glue | 60.2 | | | ARQ seasonal comparison | Steady state, Instantaneous, Continuous |
| | *Platanus occidentalis* L. | | 43.4 21.2 24.3 | | | Day-night ARQ variation | |
| | *Pistacia atlantica* Desf. | Perspex®, vacuum grease | 16.3 20 ± 8 | 12 | 29 | ARQ vs. temperature ARQ vertical transects | |
| | *Quercus calliprinos* Webb. *Malus domestica* Borkh. *Quercus ilex* L. | | | | | ARQ of incubated stem cores and leaves | |
| Ramat Hanadiv Nature Park Carmel Ridge, Israel (32.55°N, 34.94°E) | *Quercus calliprinos* Webb. | Perspex®, hot glue | 11.2 ± 1.2 | 9.8 | 12.7 | Simultaneous measurements of ARQ and shoot water potential | Steady state |
| Bartlett Experimental forest, NH, USA (44.06°N, 71.29°W) | *Acer rubrum* L. | Polypropylene [c], caulking | 20 ± 10 | 10 | 34 | ARQ seasonal comparison | Steady state |
| Harvard forest, MA, USA (42.53°N, 72.17°W) | *Acer rubrum* L. | Polypropylene, caulking | 18 ± 9 | 9 | 26 | ARQ seasonal comparison | Steady state |
| Majadas de Tiétar, Caceres, Spain (39°56'25" N, 5°46'28" W) | *Quercus ilex* L. | Perspex®, vacuum grease | 45 ± 7 | 35 | 64 | ARQ survey of 16 trees Simultaneous measurements of ARQ and sap flux density ARQ of incubated stem cores | Steady state, Instantaneous |
| Gigante peninsula, Barro Colorado Nature Monument, Republic of Panama (9°06'31" N, 79°50'37" W) | *Tetragastris panamensis* (Engl.) Kuntze | Perspex®, vacuum grease | 30.0 ± 12.5 | 12.6 | 66.8 | ARQ survey of 42 trees ARQ of incubated stem cores | Steady state, Instantaneous |
| A station of the Brazilian National Institute for Research in the Amazon (INPA), north west of Manaus, Brazil (2°38'23" S, 60°09'51") | *Scleronema micranthum* (Ducke) Ducke | Polypropylene [d] | 41.2 ± 13.3 | 27.0 | 57.8 | Seasonal comparison ARQ vertical transects In-stem ARQ and gases concentrations ARQ of incubated stem cores | Steady state, Instantaneous |


[a]All chambers were installed at ~1.3 m above the ground, except for the *Q. calliprinos* on Carmel Ridge that were
placed near to the ground due to the shrubby canopy, the low branching of the trunk and the constraint of the size
of the chamber.
[b] Chambers were made of 10 cm × 12 cm Perspex® plate with four connectors to allow attachment of sampling
flasks. The chamber on the *M. domestica* was slightly larger, 12 cm × 19 cm, with six flasks connectors. Chambers
were placed on top of a closed cell foam frame that allowed an air-tight seal between the rigid chamber and the



uneven surface of the tree stem. We used nylon straps to compress the foam, while the sealant was applied between
the foam and stem for ensuring the seal (Hilman and Angert, 2016). Sealants were silicone based vacuum grease
(Silicaid®1010 manufactured by Aidchim ltd., Raanana, Israel) or hot glue applied by a hot-glue gun.
[c] The chambers are described in (Muhr et al., 2013; Carbone et al., 2013). Briefly, the chambers were made from
an opaque plastic polypropylene pipe T-fitting with fittings for sampling flasks. Sealants were caulking (Nautiflex;
OASE GmbH, Oerel- Barchel, Germany) or hot glue applied by a hot-glue gun.
[d] Chambers were built from a 15 cm long piece of polypropylene (PP) tubing (6.5 cm OD) that was welded shut
on both sides with a PP disc (6.7 cm diameter). By cutting off a segment (height 2 cm) the tube was turned into
an incubation chamber. Opposite the chamber opening, three fittings (Sprint ESKV 20, Wiska, Germany) were
installed and sealed around the edges with liquid rubber (Dichtfix, Bindulin, Fürth, Germany). For sampling,
chambers were attached to the trees with 4 lashing straps. To achieve a gas tight seal, a frame (25 mm thick) made
from closed-porous cellular rubber (EPDM-quality, REIFF Technische Produkte GmbH, Reutlingen, Germany)
was placed between the chamber and the stem.





**Table 2** Comparison between the calculated PEPC fixation rates required to explain measured ARQ in stem cores incubations and reported PEPC fixation rate.

| | ARQ[a] ($CO_2$ efflux/$O_2$ uptake) | $O_2$ uptake[b] (nmol g.DW$^{-1}$ s$^{-1}$) | PEPC fixation rate required to explain the observed ARQ[c] (nmol $CO_2$ g.DW$^{-1}$ s$^{-1}$) | PEPC fixation rate[d] (nmol C g.DW$^{-1}$ s$^{-1}$) |
|---|---|---|---|---|
| *Quercus ilex* (n =4) | 0.44 ±0.08[c] | 3.84 ±0.30 | 2.15 | |
| *Tetragastris panamensis* (n =11) | 0.33 ±0.07 | 1.40 ±0.69 | 0.93 | |
| *Fagus sylvatica* L. | | | | 12.6 |

[a] Values are mean ±SD

[b] Dry weight (DW) was determined after drying in an oven at 60°C for two days.

[c] Calculated as $O_2$ uptake × (1-ARQ), which is an estimation of the flux of respired $CO_2$ that didn't diffused out from the core. Based on the assumption that carbohydrates with ARQ =1 are the respiratory substrate.

[d] We calculated PEPC fixation rate of *Fagus sylvatica* L. with data from Berveiller and Damesin (2008) as follow: PEPC activity (nmol C mg$^{-1}$ chl s$^{-1}$) × total chl (mg g.DW$^{-1}$) =~30 × 0.42 =12.6 nmol C g.DW$^{-1}$ s$^{-1}$

The chosen PEPC activity was the lowest among seasonal measurements.





Figure captions

Figure 1: Modeled changes in a tree stem chamber of the concentrations of $CO_2$, $O_2$, and the ratio between $\Delta CO_2$ and $\Delta O_2$, which are the changes in the gases concentrations from their initial values, and are also the difference in concentrations between the chamber and the atmosphere. The gas dynamics are based on a one-box model with

arbitrary fluxes and ARQ =0.5. The two time frames in which ARQ, the ratio of $CO_2$ efflux/$O_2$ influx, can be measured from the ratio $\Delta CO_2/\Delta O_2$ are indicated in the figure.

Figure 2: Relation between "instantaneous" ARQ (ratio of CO2 efflux/O2 influx for tree stems) measured in stem chambers after incubation of 30 minutes to a few hours and "steady state" ARQ measured in the same experiment with typically two days of incubation (n = 139). The regression forced to go through 0. The P value is the

significance of the slope estimation.

Figure 3: Summary of "steady state" apparent respiratory quotient (ARQ) measurements (ratio of CO2 efflux/O2 influx for tree stems) for 12 species (n measurements, n individuals). Gases were sampled from chambers at breast height (~1.3 m above soil surface), except for the Q. calliprinos in the Mediterranean shrubland, in which chambers were placed near the stem base due to branching stems. Vertical lines are mean values, error bars

represent one standard deviation, and colored bars represent the range of measured ARQ values. The Peru data is after Angert et al. (2012). The horizontal bars were ordered according to increasing mean ARQ.

Figure 4: Seasonal dynamics of "steady state" apparent respiratory quotient (ARQ, the ratio of $CO_2$ efflux/$O_2$ influx for tree stems) of five individual trees from five different species. Phenology stage index determined according to: "Defoliation"- from beginning of autumn color development to the end of the fall, "Winter

dormancy"- when the tree was bare from leaves, "Leaf regeneration"- from bud burst to early leaf development stage. The *Q. calliprinos* is evergreen. Markers are mean values and error bars are SD of duplicate samples from the same stem chamber. Markers connected with solid lines represent measurements with chambers at breast height (~1.3 m above soil surface). Markers connected with dashed lines represent measurements with chambers positioned at the stem base. The trees grew on Hebrew University campus in Jerusalem, Israel.

Figure 5: Instantaneous apparent respiratory quotient (ARQ, ratio $CO_2$ efflux/$O_2$ influx of a stem, ± SD of duplicates) values measured over a day-night-day transition in Jerusalem, Israel in July 2012 (a) and April 2013 (b) from different trees growing on Hebrew University campus in Jerusalem, Israel. *Q. calliprinos* was measured at two different heights on the stem. First sampling was taken during daylight (day 1), next sampling before dawn (pre-dawn) and last sampling during daylight of the successive day (day 2).

Figure 6: Diurnal patterns of (a) $O_2$ influx to the stem and $CO_2$ efflux from the stem, (b) chamber temperature, and (c) instantaneous apparent respiratory quotient (ARQ, ratio $CO_2$ efflux/$O_2$ influx for tree stems). Shaded areas



ndicate night periods. Error bars are 95% confidence bounds. All data were obtained from a single *M. domestica* tree during 24-28 April 2013 on Hebrew University campus in Jerusalem, Israel.

Figure 7: Instantaneous apparent respiratory quotient (ARQ, ratio of $CO_2$ efflux/$O_2$ influx for tree stems) measured from stem chambers installed at different heights above the ground on a *S. micranthum* tree in Brazil. At the same heights ARQ was measured from 4 cm in-stem probes. The measurements were conducted during 30 March and 18 October 2012. Error bars represents SD of duplicate samples from the same stem chamber.

Figure 8: Comparisons of stem chamber apparent respiratory quotient (ARQ, ratio $CO_2$ efflux/$O_2$ influx for tree stems, "steady state") to ARQ measured from incubations of stem cores (ratio $CO_2$ increase/$O_2$ decrease), by species (n individuals) in different sites. Values are means ±SD.

Figure 9: (a) $O_2$ uptake rate (nmol g.FW$^{-1}$ s$^{-1}$) and (b) apparent respiratory quotient (ARQ, ratio $CO_2$ increase/$O_2$ decrease) of *Q. ilex* leaves and stem cores incubated in a closed system (n =4). Values are means ± SD. Asterisks indicate significant difference between tissues at each time step (* $P < 0.05$, ** $P < 0.01$, *** $P < 0.0001$ in Student's t-test). Different letters indicate significant difference in Tukey-Cramer HSD analysis that followed one-way analysis of variance (ANOVA) within tissue type, between time steps.





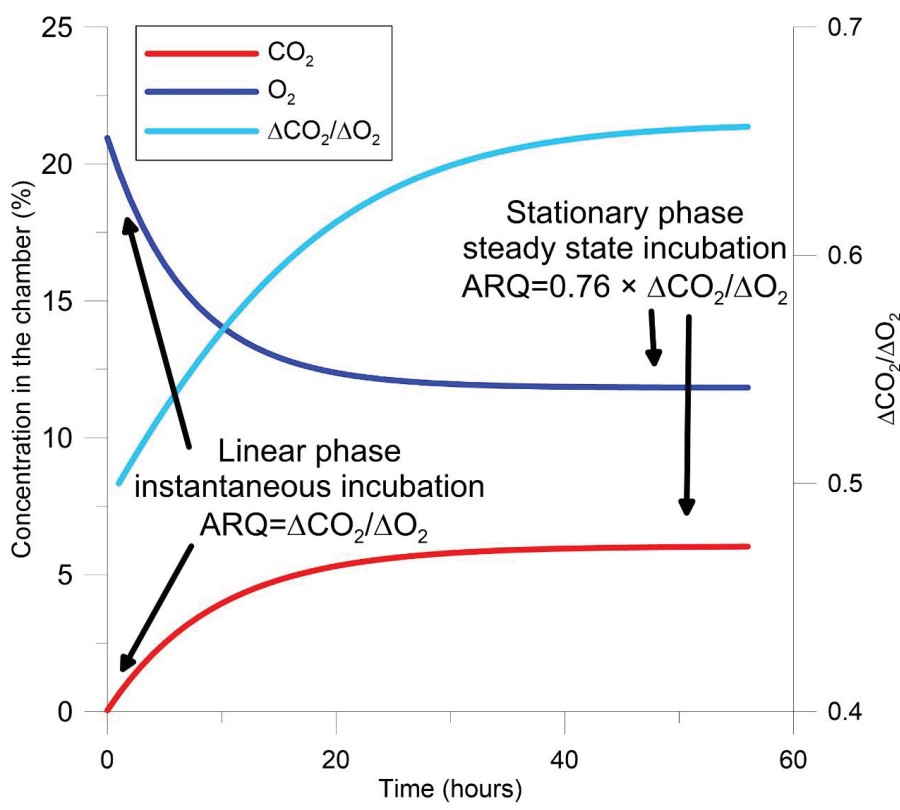

10    Figure 1





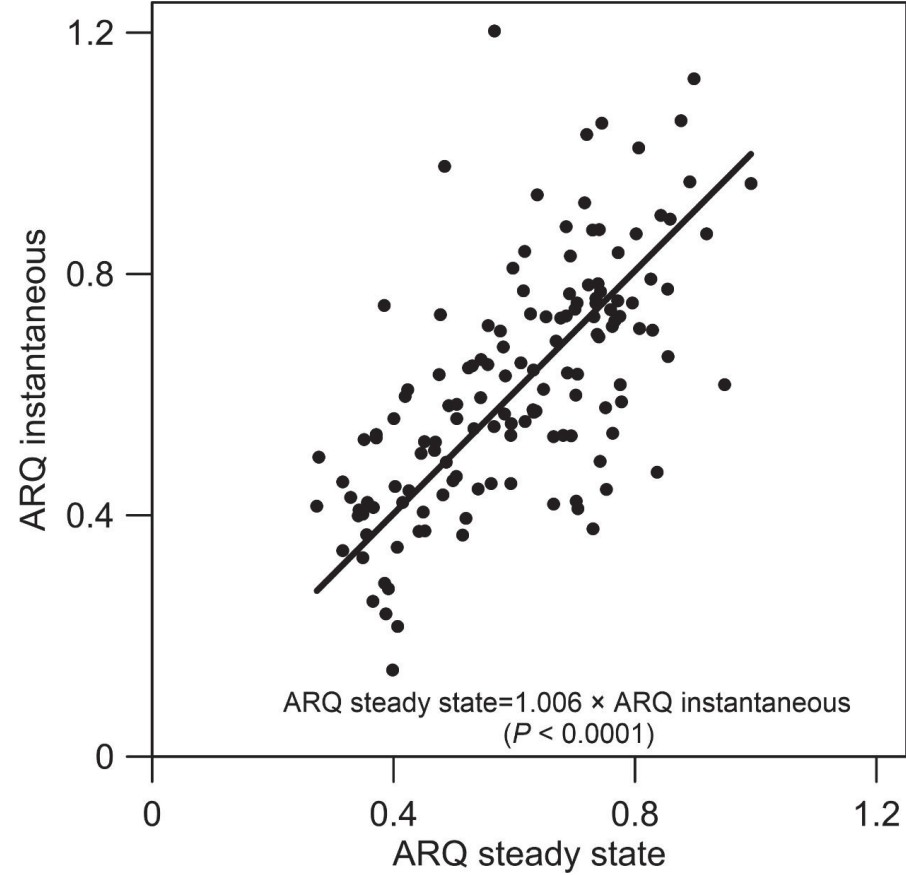

Figure 2





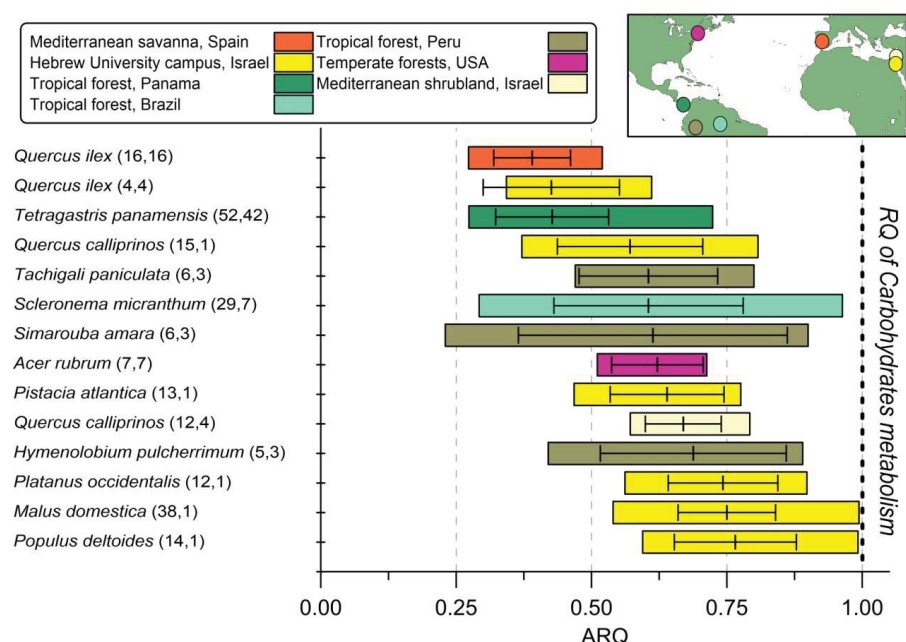

Figure 3



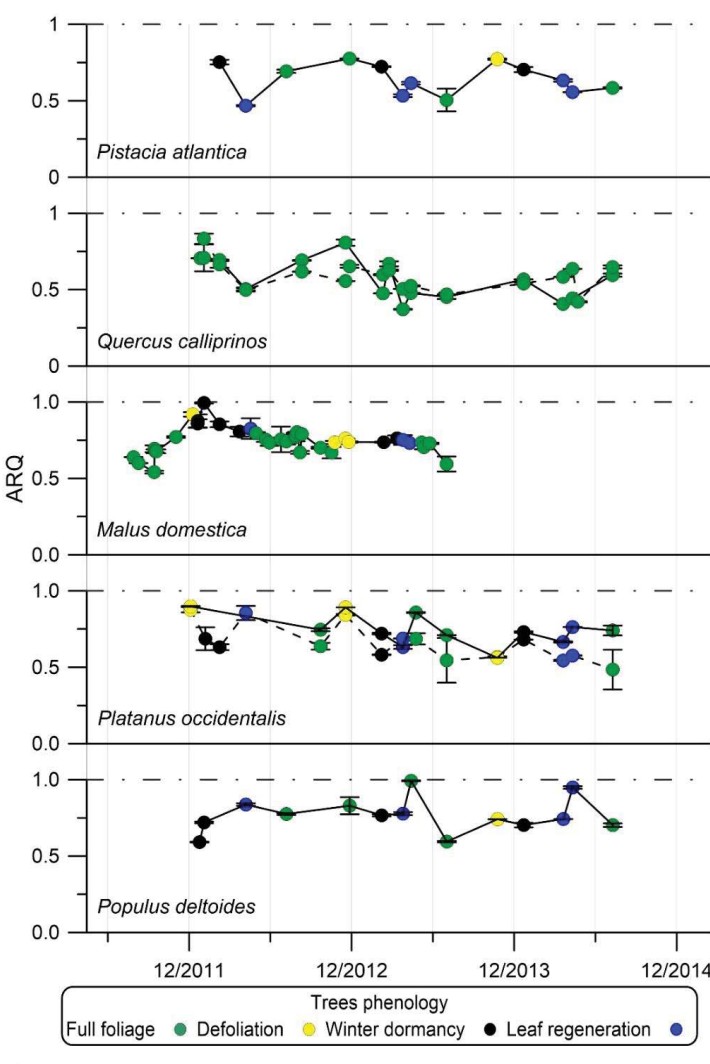

Figure 4





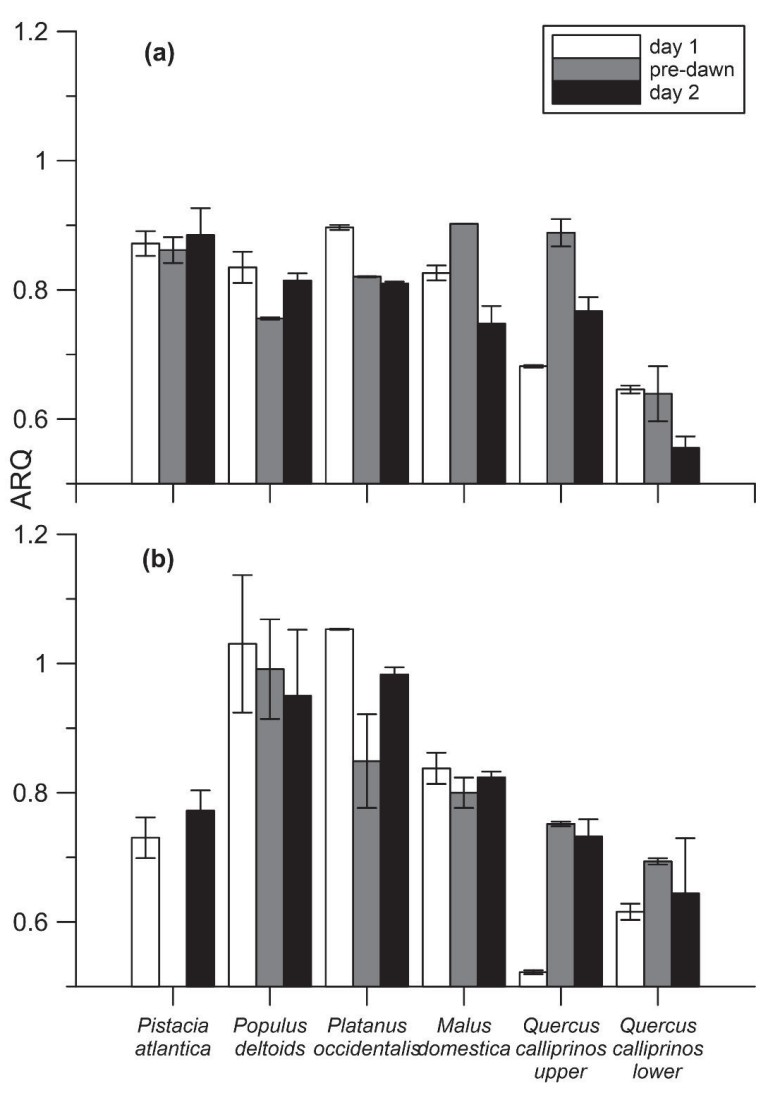

Figure 5





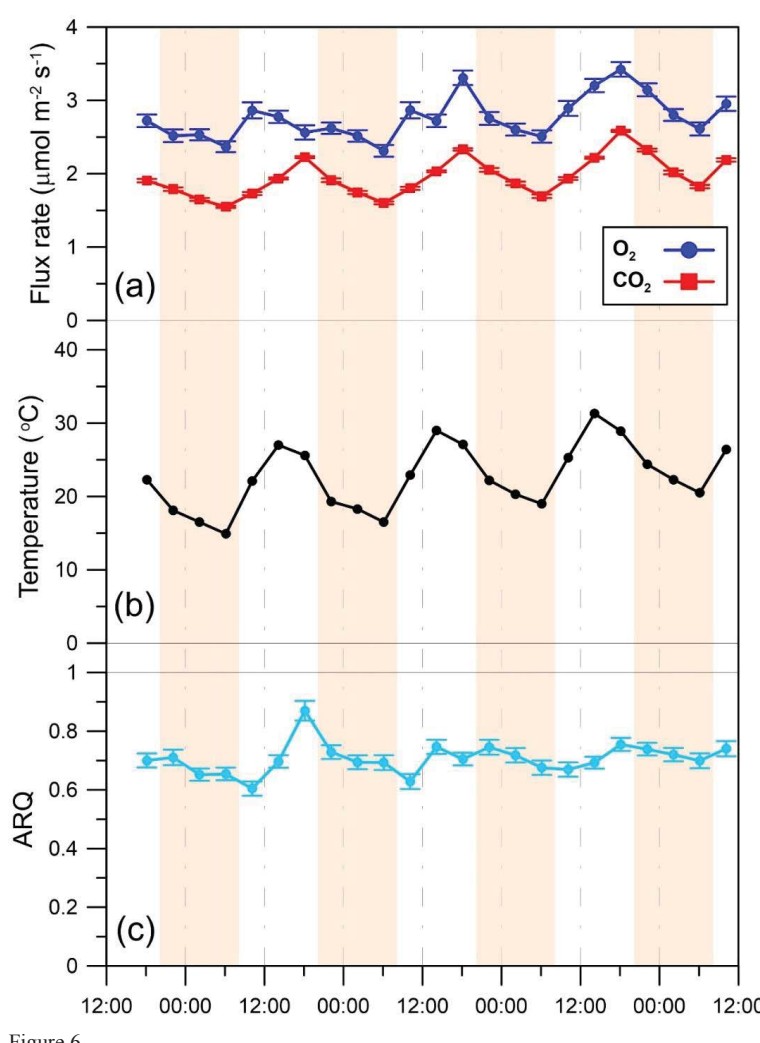

5    Figure 6





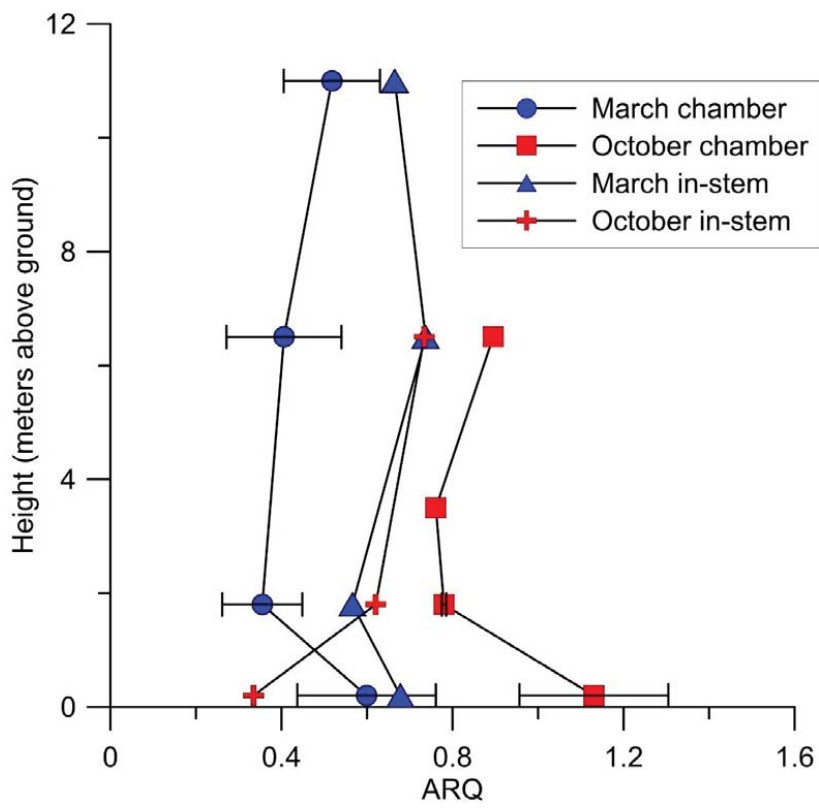

Figure 7



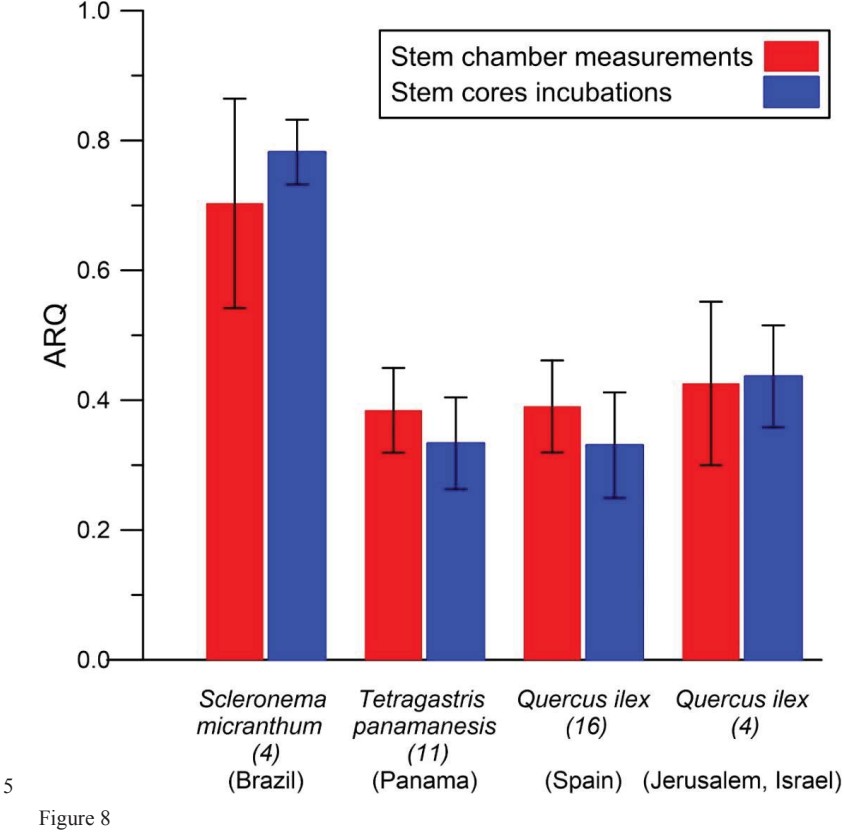

Figure 8



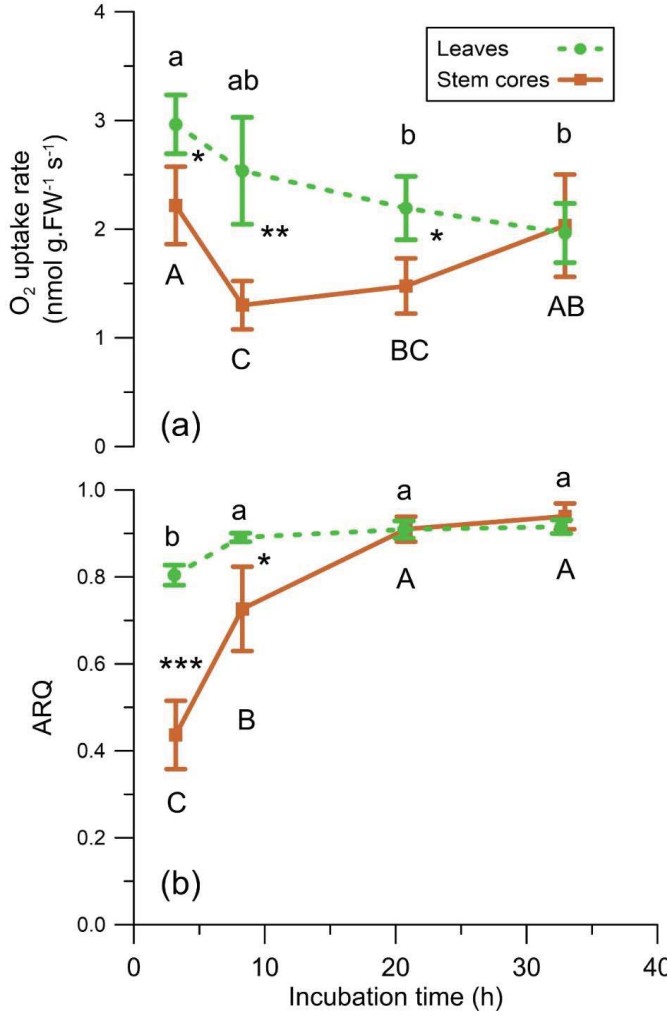

Figure 9