# Peer review of "Comparison of CO2 and O2 fluxes demonstrate retention of respired CO2 in tree stems from a range of tree species"

_Biogeosciences, 2018_

## Referee Comment (RC1) · Anonymous Referee #1 · 18 Sep 2018

This article presents data on a range of tree species that demonstrates the apparent respiratory quotient measured on stems (ARQ, the ratio of CO2 emission and O2 uptake) is often well below a value of $\sim$1 expected for aerobic metabolism of carbohydrates. The authors do an excellent job outlining possible explanations for this discrepancy in the introduction and discussion. They pull together an analysis of multiple experiments, including field measurements and lab incubations of samples from 12 species across half a dozen sites. While the methods are a bit confusing due to integration across so many disparate experiments, overall they are clear. Of particular interest in the lab incubations of excised stem and leaf material in the discussion of transport of DIC.

[Figure]

The biggest potential issue with this manuscript is the choice of ANOVA as an analytical approach, particularly in the use of ARQ as a response variable. Since ARQ is not measured directly, but is the ratio of the two measurements, it does not necessarily have the correct statistical properties for ANOVA. In particular, if measurement error scales with the component fluxes of ARQ, then the relative error of ARQ increases as the component fluxes decrease. This is of particular concern in light of the admission that in multiple cases, samples were excluded for having very low flux measurements, presumably approaching detection limits. I would recommend the authors consider another approach that preserves the original scaling of the measurements, such as ANCOVA or multiple regression. I would not object to summarizing some of the findings in terms of ARQ, however, as it is a useful tool for explanation.

Another potential issue is the combination of analyses across so many species and sites. While, on one hand, this a strength of the paper, there is an implicit assumption that the mechanisms are consistent across species and sites. This is not necessarily the case, especially in regard to the transport of DIC in sap, which could be effected greatly by species or wood anatomy. One may expect that this process and the contribution of transport to observed ARQ values would depend greatly on the depth of active sapwood, vessel size and other anatomical characteristics, such as medullary rays.

Overall, the basic observation of ARQ being less than unity across many studies is a useful contribution to the literature and the discussion of the topic is well written and clear. One possible spot for improvement is connecting the putative mechanism of PEPC fixation of $CO_2$ with transport and canopy-level measurements of $CO_2$ and $O_2$ exchange. In particular how this could result in similar decoupling of the component fluxes in time rather than or in addition to spatial decoupling from processes such as the transport of DIC in sap.

---

## Author Comment (AC1) · 27 Sep 2018

Thank you for your comments. Major concerns were raised and addressed below.

The question of appropriateness of ANOVA tests for ARQ data. The perception that relative measurement errors increase in smaller signals is correct. However, we used ANOVA for compare measurements in which the gas fluxes were in similar magnitude, thus the supposed analytical errors are similar. In addition, our instrumental design minimizes analytical errors. Most of the ARQ results in the paper, including those ANOVA was applied with, were measured in the Hampadah, a closed-system contains both CO2 and O2 analyzers. The inclusion of both analyzers in the same system means that temperature and pressure effects will occur simultaneously, and will be canceled while dividing the CO2/O2 measurements. Thus, no systematic bias in the ARQ estimation is expected. It was demonstrated in very good precision for duplicate ARQ samples ($\pm$0.01) (Hilman and Angert, 2016). The O2 measurement with the Hampadah is validated by measurement of O2 by mass spectrometry; Licor and optode measurements are also calibrated to ensure accuracy (Hilman and Angert, 2016). Yet, the precision in the gases concentrations is poorer than for ARQ and the error for the fluxes are higher than for the ARQ. Therefore, the ARQ measurement can be regarded as direct measurement thus compatible with ANOVA. As results are measured for multiple trees, the most important expression of uncertainty is how reproducible the measurement is - and how does that standard deviation across measurements compare to the expressed error for an individual measurement.

Regarding the comment about possible different mechanisms in different stem anatomies - we are not refuting that. In the discussion, L401-404, we mentioned that the ARQ value can be the result of combination of effects. In the revised paper we would further highlight that the mechanisms responsible for a given ARQ value may vary between species and wood anatomies.

Regarding your last point about connecting the stem measurements and possible PEPC fixation to canopy-level CO2 and O2 measurements, we would be happy to include discussion of this in a revised paper. We have additional unpublished results of simple labeling experiments that demonstrate dark fixation in 1-2 species, providing confirmation to PEPC activity. However, the other comment of the reviewer is that this paper already is complex because it includes results collected at multiple sites and multiple times. If the editor/reviewer prefer we can include such discussion and also the additional results in the revision.

Hilman, B., and Angert, A.: Measuring the ratio of CO2 efflux to O2 influx in tree stem respiration, Tree Physiology, 36, 1422-1431, 10.1093/treephys/tpw057, 2016.

---

## Referee Comment (RC2) · Anonymous Referee #2 · 12 Oct 2018

This manuscript describes the results of a series of field experiments to measure the Apparent Respiratory Quotient (ARQ) and its influencing factors in 9 different tree species. Sites were located in Peru, Brazil, Panama, Israel, Spain, and the US. The authors sought to obtain estimates of ARQ from trees in different biomes and across seasons, and to test whether ARQ varies with xylem stream characteristics and tree height.

The main points as I understand them are as follows: 1) Theoretically, ARQ should be near 1 in tree stems utilizing carbohydrates as a respiratory substrate. The authors found that ARQ was substantially less than 1 in all trees measured, with a range of

[Figure]

ARQ from 0.39 to 0.78. Previous studies in tropical trees also found ARQ values less than 1, which suggested retention of respired carbon within the stem. With this paper, the authors extend previous results and show that ARQ values <1 are common in trees across biomes and across seasons.

2) The authors also found that hypothesized explanations for ARQ values <1 related to Co2 dissolution and transport as the main Co2 export mechanism were generally insufficient to explain the low ARQ values observed. Specifically, they found that sap flow was not correlated with ARQ, whereas a negative correlation would be expected in the case of Co2 transport in the stem. Additionally, temperature controls on ARQ are not sufficient to explain the deviation from unity.

This paper has an important result, which is that ARQ values lower than 1 are widespread across biomes, and these low values cannot be explained by dissolution and transport of respired CO2 in the xylem stream. Overall, while well-written overall, this paper has significant issues with organization and clarity. The introduction is compelling and reads smoothly, but the methods section in particular is difficult to follow. Additionally, the discussion section introduces a new concept to explain the results and dwells on concepts that the introduction stated were not important. There are different methods used and experiments performed at each study site, and this information is not presented logically. Table 1 was very difficult to read due to spacing within the table and it did not contain easily obtainable information about which experiments were performed at which site. I recommend reorganization of these sections. Additionally, I was unsatisfied with the PEPC explanation for the low ARQ values that was only introduced at the very end of the paper. This new concept was introduced without sufficient context, such as 'what is a reasonable value for PEPC fixation?' – there is only 1 true value presented in the table. This concept should be introduced earlier in the paper with proper setup, because as is it feels like a surprise. Finally, I am unconvinced by the authors' assertion that ARQ values from "instantaneous" vs. "steady state" sampling are equivalent because the regression was forced through zero and R2 was not

reported, masking bias that could be present based on previous studies. The authors achieve their original objective to obtain estimates of ARQ in different biomes across seasons, but this important result was obfuscated by a complicated methods section and a scattered discussion section. I recommend significant revisions to this paper that include: reorganization of methods and results sections for clarity, making the discussion mirror the introduction, and improving overall cohesiveness. The authors lay out clear objectives in the introduction, but the discussion has a lot of information in it that isn't set up in the introduction. To make this story more cohesive, the authors should keep their main objectives in mind in the revisions, introduce important concepts earlier in the paper, and make sure the discussion section follows logically from the results presented.

Major comments: I recommend reorganization and attention to consistency in referring to species vs. sites. For example, Figure 4 refers to species, but when presented in the results (lines 260-263), the "Bartlett" and "Harvard" are referenced. The reader should not have to go to Table 1 for reference to understand to which panel in figure 4 the text refers. Sometimes the authors mix species and site names in the same sites, for example in lines 317-318 when they refer to trees in Jerusalem. The reader should again, not have to refer to Table 1 to figure out which species were in Jerusalem.

Site names should be consistent throughout the manuscript. Sometimes sites are referred to by name (e.g. "Hebrew University" or "Carmel Ridge"), sometimes by location (e.g. "Jerusalem" or "Brazil", and sometimes by a more general name like "Israel"). In this example, there is no "Hebrew University" referred to in Table 1 so the reader cannot even be certain which site is being discussed when this term is used. Please be consistent throughout the manuscript with your names for each site.

You assert that ARQ values from "instantaneous" vs. "steady state" sampling are equivalent – however, the regression was forced through zero, and authors only report the slope. Forcing the regression through the origin will mask any main effect bias. There is no reason to assume the regression will go through zero, conversely, Angert et al.

(2012) showed a large difference in ARQ between these two approaches (Figure 2). You state several times that the methods are equivalent and that there are no measurement effects, but the test used to support the statement is insufficient.

I suggest reorganizing the "measurements made in the site" column of table 1 so it is easier to understand. It would be better if the reader could look at the numbered list of experiments (lines 182-191) and know which experiment was performed at which site. Please have the measurements in the table use the same wording as the subsections in the methods section. For example, the reader should be able to read the section heading 3.3 "Tissue Incubations" and easily discern from Table 1 where these measurements were performed.

The methods section (in particular sections 2.2 – 2.6) is difficult to follow and should be reorganized for clarity. For example, you could try organizing section 2.2 by site, which might make it easier to keep track of which experiment was performed at which site. Or, you could try incorporating the numbered list of experiments (lines 182-191) in the following sections. As is, the section is difficult to follow.

Minor Comments: I found the lengthy calculation in the discussion section (lines 383-393) to be strange. Again, it is nearly the end of the paper, and a new two pool model is introduced and a calculation is performed. Please clarify the purpose of this calculation to answering your overall objectives, and consider how to shorten and make it more conceptual.

The distinction in greenness between Harvard and Bartlett forest is not discernable from Figure 4 (discussed lines 320-322), please clarify.

I was surprised that pre-dawn water potentials were not referred to as a measure of water stress in lines 190-192, especially since stress is stated as a potential explanation for lower ARQ values in the discussion (line 307-308).

Similarly, I was surprised to see the lengthy discussion of lipid storage in stems in lines

304 to 312, when the authors discount this as a reasonable explanation for lower ARQ values in the introduction in lines 39-41. The introduction made it seem like this was an unlikely explanation anyway, as only a few genera of tree species store lipids in tree stems.

Why does ARQ plateau at 0.7 in the model presented in Figure 1? Shouldn't it plateau closer to 1 if that's what the theory suggests?

Table 1 is very hard to read due to the spacing of text

Figure 3 is graph is good and easy to read, but the two yellow colors on this graph are difficult to distinguish

Figure 4: It is difficult to distinguish colors and symbols on these graphs as many points overlap. Perhaps different symbols for the different heights would help?

Post-hoc comparisons – your methods (lines 249-251) state you did Tukey's post hoc comparisons, but I don't see letters corresponding to the post hoc tests like I'd expect to see on Figures 8 and 5 in particular. Were these tests performed for the corresponding analyses, and if so, why aren't the results on the figure?

Line by line comments: Line 275 – what is the duplicates error?

Please be consistent with the placement of the ARQ measurement type in-text, for example in line 188, "continuous" should be right after "ARQ" instead of at the end of the sentence.

Line 342 should read "must" not "much".

---

## Author Comment (AC2) · 27 Oct 2018

Thank you for your comments. We addressed the major concerns below.

Concerns were raised regarding the organization of the paper. The paper presents a synthesis of results from different sites with different experimental designs and methods, and we made big efforts to report this synthesis in fluent and coherence manner. Yet, from the reviewer comments there is more room for improvement. In the revised paper we will follow editing suggestions provided by the reviewer: re-organizing methods section and table 1, broadening the introduction for PEPC as a potential mechanism. We will also edit the discussion to mirror the introduction, improving overall

[Figure]

cohesiveness. We also will take care to keep sites names consistent. Obviously, we will clarify/correct minor issues the reviewer pointed out.

Steady state vs instantaneous ARQ determination. The reviewer is concerned about the equivalency of the steady-state and the instantaneous measurements as determined from the linear regression (Fig 2) and from the results of Angert et al. (2012). Specifically, the reviewer criticizes the decision to force the regression line in figure 2 through the origin. Figure 2 presents a linear regression with a slope of ~1 (but with considerable scattering) where the variables are the steady-state and the instantaneous ARQ measurements of the same incubations. We aimed to demonstrate by that the overall validity of our box-model that predicts same ARQ values shortly after the beginning of chamber incubation (instantaneous) and after steady-state is reached (>day). As we discussed in the paper and previously (Hilman & Angert, 2016), the considerable scattering around perfect agreement can reflect several things: (1) real differences in the ability of the measurement method or (2) real temporal differences in ARQ that occur between the time the instantaneous and steady-state measurements are realized (the model we use assumes constant ARQ with time). Examples for temporal changes in ARQ are shown in Fig. 5-6. The reviewers' comment has led us to re-think whether linear regression is the best way to demonstrate the adequacy of the steady-state and the instantaneous measurements, because there are not necessarily dependent-independent variables. Instead, we think a better way to illustrate the agreement is to plot the data points, the 1:1 slope, and to report the mean difference (0.02) and RMSD between determinations which is 0.15.

Why does ARQ plateau at 0.7 in the model presented in Figure 1? Shouldn't it plateau closer to 1 if that's what the theory suggests? The theory dictates the ratio DCO2/DO2 in the beginning of stem incubation should equal 0.76*DCO2/DO2 at the plateau stage. In this model run ARQ=0.5, hence the DCO2/DO2 plateaus at ~0.65. Post-hoc comparisons. We performed post-hoc comparison only in the experiment presented in Figure 9, where the letters indicate significant differences are shown. Line by line

comments: Line 275 – what is the duplicates error? Single ARQ measurement is the average of duplicate flasks taken from the stem chamber, and the error is the standard deviation. We will clarify this in the text. I was surprised that pre-dawn water potentials were not referred to as a measure of water stress in lines 190-192, especially since stress is stated as a potential explanation for lower ARQ values in the discussion (line 307-308). Drought treatment was mentioned in these lines as one potential explanation for lipid respiration that could cause a decrease in ARQ. As natural trees in Israel are acclimated to the long dry Mediterranean summer, we are not sure the term "stress" describes exactly the water status of these trees. For eliminate misunderstanding we will remove the reference to "stress" in the text.

Angert, A., Muhr, J., Negron-Juarez, R., Alegria-Muñoz, W., Kraemer, G., Ramirez-Santillan, J., . . . Trumbore, S. E. (2012). Internal respiration of Amazon tree stems greatly exceeds external CO2 efflux. Biogeosciences, 9, 4979-4991. Hilman, B., & Angert, A. (2016). Measuring the ratio of CO2 efflux to O2 influx in tree stem respiration. Tree Physiology, 36(11), 1422-1431. doi:10.1093/treephys/tpw057

---

## Author Response (AR1)

Response to reviewers

**"Comparison of $CO_2$ and $O_2$ fluxes demonstrate retention of respired $CO_2$ in tree stems from a range of tree species"**

**Hilman et al. reply**

We thank the reviewers for their comments. Detailed responses to the comments appear below, numbered and in **bold**. The main changes were: (1) Re-organization of the methods section and Table 1; (2) adding information about PEPC and the possible fates of the refixed carbon to the introduction and discussion; (3) Re-organization of the discussion, especially its second half, and including a short discussion on the possible effect of corticular photosynthesis (L435-441).

**Reviewer 1**

The biggest potential issue with this manuscript is the choice of ANOVA as an analytical approach, particularly in the use of ARQ as a response variable. Since ARQ is not measured directly, but is the ratio of the two measurements, it does not necessarily have the correct statistical properties for ANOVA. In particular, if measurement error scales with the component fluxes of ARQ, then the relative error of ARQ increases as the component fluxes decrease. This is of particular concern in light of the admission that in multiple cases, samples were excluded for having very low flux measurements, presumably approaching detection limits. I would recommend the authors consider another approach that preserves the original scaling of the measurements, such as ANCOVA or multiple regression. I would not object to summarizing some of the findings in terms of ARQ, however, as it is a useful tool for explanation.

1. **The perception that relative measurement errors increase in smaller signals is correct. However, we used ANOVA for compare measurements in which the gas fluxes were similar in magnitude, thus the relative analytical errors are of similar magnitude. In addition, our instrumental design minimizes analytical errors. Most of the ARQ results in the paper, including those ANOVA was applied with, were measured in the *Hampadah,* a closed-system that contains both $CO_2$ and $O_2$ analyzers. The inclusion of both analyzers in the same system means that temperature and pressure effects will occur simultaneously, and will be canceled while dividing the $CO_2/O_2$ measurements. Thus, no systematic bias in the ARQ estimation is expected. This was demonstrated in very good precision for duplicate ARQ samples (±0.01) (Hilman and Angert, 2016a). The $O_2$ measurement with the *Hampadah* is validated by measurement of $O_2$ by mass spectrometry; Licor and optode measurements are also calibrated to ensure accuracy (Hilman and Angert, 2016a). Yet, the precision in the concentrations of the gases is poorer than for ARQ and the errors for the individual gas fluxes are higher than for the ARQ. Therefore, the ARQ measurement can be regarded as a direct measurement of the ratio and thus compatible with ANOVA. As results are measured for multiple trees, the most important expression of uncertainty is how reproducible the measurement is — and how does that standard deviation across measurements compare to the expressed error for an individual measurement.**

Another potential issue is the combination of analyses across so many species and sites. While, on one hand, this a strength of the paper, there is an implicit assumption that the mechanisms are consistent across species and sites. This is not necessarily the case, especially in regard to the transport of DIC in sap, which could be effected greatly by species or wood anatomy. One may expect that this process and the contribution of transport to observed ARQ values would depend greatly on the depth of active sapwood, vessel size and other anatomical characteristics, such as medullary rays.

2. **The text was corrected according to this comment. In the revised discussion we linked wood anatomy to the contradicting literature regarding the contribution of in-stem $CO_2$ to surface efflux (L401-L411). Later in the discussion we further highlight that the ARQ value is probably the sum of numerous mechanisms, which might vary between species and wood anatomies (L432-435 and L441-443).**

One possible spot for improvement is connecting the putative mechanism of PEPC fixation of $CO_2$ with transport and canopy-level measurements of $CO_2$ and $O_2$ exchange. In particular how this could result in similar decoupling of the component fluxes in time rather than or in addition to spatial decoupling from processes such as the transport of DIC in sap.

3. **We added discussion in this regard. In L422-431 we offer two sinks for PEPC products: export of malate to the canopy where 'C4-like photosynthesis' might happen, and/or export of organic acids to the soil as root exudates. In L449-454 we predict the $CO_2$ and $O_2$ exchange expected for each of those possible sinks: ARQ >1.0 in the rhizosphere as result of organic acid catabolism, and increase of the photosynthetic oxidative ratio ($O_2$ produced/$CO_2$ consumed) as the internally transported C replaces the atmospheric $CO_2$ when assimilation is measured.**

**Reviewer 2**

This paper has an important result, which is that ARQ values lower than 1 are widespread across biomes, and these low values cannot be explained by dissolution and transport of respired CO2 in the xylem stream. Overall, while well-written overall, this paper has significant issues with organization and clarity. The introduction is compelling and reads smoothly, but the methods section in particular is difficult to follow. Additionally, the discussion section introduces a new concept to explain the results and dwells on concepts that the introduction stated were not important. There are different methods used and experiments performed at each study site, and this information is not presented logically. Table 1 was very difficult to read due to spacing within the table and it did not contain easily obtainable information about which experiments were performed at which site. I recommend reorganization of these sections.

4. **The methods section was revised as well as Table 1 according to the reviewer's suggestions. We edited section 2.2 (L164-220) by providing a list of the conducted experiments, and which questions those experiments test. The list is linked now to Table 1, which was re-designed.**

Additionally, I was unsatisfied with the PEPC explanation for the low ARQ values that was only introduced at the very end of the paper. This new concept was introduced without sufficient context, such as 'what is a reasonable value for PEPC fixation?' – there is only 1 true value presented in the table.

This concept should be introduced earlier in the paper with proper setup, because as is it feels like a surprise.

5. **An introduction to PEPC was added (L70-74), mirroring the discussion. Unfortunately, PEPC measurements in stems are extremely rare. We were able to find one more paper that enables calculation of PEPC activity that is comparable to our measurements (Ivanov et al., 2005). The calculation is included in Table 2 and results in activity similar to that in the first cited paper (Berveiller and Damesin, 2008).**

Finally, I am unconvinced by the authors' assertion that ARQ values from "instantaneous" vs. "steady state" sampling are equivalent because the regression was forced through zero and R2 was not reported, masking bias that could be present based on previous studies. The authors achieve their original objective to obtain estimates of ARQ in different biomes across seasons, but this important result was obfuscated by a complicated methods section and a scattered discussion section.

6. **The aim of Fig. 2 is to demonstrate the overall validity of this comparison.  Our box-model allows comparison of ARQ values measured shortly after the beginning of chamber incubation (instantaneous) and after steady-state is reached (>day). Considering the reviewers' comment, we agree that linear regression between the ARQ determined each way is not the best way to demonstrate the adequacy of the steady-state and the instantaneous measurements, because they are not dependent-independent variables. The figure was slightly changed, and the 1:1 line (which is what we expect if the methods agree) was plotted instead of the linear fit. We additionally report the mean difference and RMSD from the 1:1 line, which are respectively 0.02 and 0.15. As we discussed in the paper (L349-358) and previously (Hilman and Angert, 2016a), the considerable scatter around the 1:1 line and the large RMSD could partly be attributed to temporal differences in ARQ during the time between the instantaneous and steady-state measurements, while the model assumes constant ARQ with time. Additionally, the precision for "instantaneous" ARQ was lower than for "steady state" values, due to smaller changes in $O_2$ over the shorter time periods. This may also contribute to the scatter in Fig. 2 (Hilman and Angert, 2016b).**

I recommend significant revisions to this paper that include: reorganization of methods and results sections for clarity, making the discussion mirror the introduction, and improving overall cohesiveness. The authors lay out clear objectives in the introduction, but the discussion has a lot of information in it that isn't set up in the introduction. To make this story more cohesive, the authors should keep their main objectives in mind in the revisions, introduce important concepts earlier in the paper, and make sure the discussion section follows logically from the results presented.

7. **Please see answers 4-6.**

Major comments: I recommend reorganization and attention to consistency in referring to species vs. sites. For example, Figure 4 refers to species, but when presented in the results (lines 260-263), the "Bartlett" and "Harvard" are referenced. The reader should not have to go to Table 1 for reference to understand to which panel in figure 4 the text refers. Sometimes the authors mix species and site names in the same sites, for example in lines 317-318 when they refer to trees in Jerusalem. The reader should again, not have to refer to Table 1 to figure out which species were in Jerusalem. Site names should be consistent throughout the manuscript. Sometimes sites are referred to by name (e.g. "Hebrew University" or "Carmel Ridge"), sometimes by location (e.g. "Jerusalem" or "Brazil", and sometimes by a more general name like "Israel"). In this example, there is no "Hebrew University" referred to in Table 1 so the reader cannot even be certain which site is being discussed when this term is used. Please be consistent throughout the manuscript with your names for each site.

8. **The study sites names are consistent now throughout the manuscript.**

You assert that ARQ values from "instantaneous" vs. "steady state" sampling are equivalent – however, the regression was forced through zero, and authors only report the slope. Forcing the regression through the origin will mask any main effect bias. There is no reason to assume the regression will go through zero, conversely, Angert et al. (2012) showed a large difference in ARQ between these two approaches (Figure 2). You state several times that the methods are equivalent and that there are no measurement effects, but the test used to support the statement is insufficient.

9. **Please see answer 6.**

I suggest reorganizing the "measurements made in the site" column of table 1 so it is easier to understand. It would be better if the reader could look at the numbered list of experiments (lines 182-191) and know which experiment was performed at which site. Please have the measurements in the table use the same wording as the subsections in the methods section. For example, the reader should be able to read the section heading 3.3 "Tissue Incubations" and easily discern from Table 1 where these measurements were performed.

10. **Please see answer 4. In addition, we reorganize the column, which is named now "Experiments in the site" and contains information about the experiments done in the site (with reference to the list in section 2.2) and the dates the experiments were performed. The "tissue incubations" (sections 2.6 and 3.3) are referred in the Table according to the list in section 2.2 (G) and specifically by the measured tissue (stem cores/leaves).**

The methods section (in particular sections 2.2 – 2.6) is difficult to follow and should be reorganized for clarity. For example, you could try organizing section 2.2 by site, which might make it easier to keep track of which experiment was performed at which site. Or, you could try incorporating the numbered list of experiments (lines 182-191) in the following sections. As is, the section is difficult to follow.

11. **Was corrected, please see answers 4, 10. We organized the experiments in section 2.2 by experiments and not by site.**

Minor Comments: I found the lengthy calculation in the discussion section (lines 383- 393) to be strange. Again, it is nearly the end of the paper, and a new two pool model is introduced and a calculation is performed. Please clarify the purpose of this calculation to answering your overall objectives, and consider how to shorten and make it more conceptual.

**12. The calculation using the two-pool mixing model was removed from the text.**

The distinction in greenness between Harvard and Bartlett forest is not discernable from Figure 4 (discussed lines 320-322), please clarify.

**13. Figure 4 presents trees measured in the Jerusalem site. The results from Harvard and Bartlett forests are not presented visually.**

I was surprised that pre-dawn water potentials were not referred to as a measure of water stress in lines 190-192, especially since stress is stated as a potential explanation for lower ARQ values in the discussion (line 307-308).

**14. Stress was mentioned in these lines as potential explanation for lipid respiration that can cause a decrease in ARQ. As natural trees in Israel are acclimated to the long dry Mediterranean summer, we don't think the term "stress" describes exactly the water status of these trees. To eliminate misunderstanding we removed the reference to "stress" in the text.**

Similarly, I was surprised to see the lengthy discussion of lipid storage in stems in lines 304 to 312, when the authors discount this as a reasonable explanation for lower ARQ values in the introduction in lines 39-41. The introduction made it seem like this was an unlikely explanation anyway, as only a few genera of tree species store lipids in tree stems.

**15. We revised the description in the introduction (L40-41) and edited the discussion about lipid storage (L329-333). In brief, lipids are believed to have small importance in storage and respiration in most trees. However, this assumption is based on very few lipid measurements in the literature, and we could not confidently refute the possibility that lipids potentially play an important role in tree stems.**

Why does ARQ plateau at 0.7 in the model presented in Figure 1? Shouldn't it plateau closer to 1 if that's what the theory suggests?

**16. The theory dictates the ratio DCO2/DO2 in the beginning of stem incubation should equal 0.76*DCO2/DO2 at the plateau stage. In this model run, ARQ=0.5, hence the DCO2/DO2 plateaus at ~0.65.**

Figure 3 is graph is good and easy to read, but the two yellow colors on this graph are difficult to distinguish

**17. Was corrected.**

Figure 4: It is difficult to distinguish colors and symbols on these graphs as many points overlap. Perhaps different symbols for the different heights would help?

**18. The symbols of the stem-base chambers are smaller now. In the software used to design the plots it is impossible in this type of plot to change the symbol style.**

Post-hoc comparisons – your methods (lines 249-251) state you did Tukey's post hoc comparisons, but I don't see letters corresponding to the post hoc tests like I'd expect to see on Figures 8 and 5 in particular. Were these tests performed for the corresponding analyses, and if so, why aren't the results on the figure?

19. **We performed post-hoc comparison only in the experiment presented in figure 9, where the letters indicate significant differences are shown.**

Line by line comments: Line 275 – what is the duplicates error?

20. **Single ARQ measurement is the average of duplicate flasks taken from the stem chamber, and the error is the standard deviation (SD). This is now clarified it in the text (L140-141).**

Please be consistent with the placement of the ARQ measurement type in-text, for example in line 188, "continuous" should be right after "ARQ" instead of at the end of the sentence. Line 342 should read "must" not "much".

21. **Was corrected.**

[revised manuscript text omitted]

Figure 1

[Figure]

ARQ steady state=1.006 × ARQ instantaneous
(*P* < 0.0001)

[Figure]

Figure2

[Figure]

[Figure]

Figure 3

[Figure]

[Figure]

Figure 4

[Figure]

Figure 5

[Figure]

Figure 6

[Figure]

Figure 7

[Figure]

Figure 8

[Figure]

Figure 9